# The Ethyl Acetate Extract of *Phyllanthus emblica* L. Alleviates Diabetic Nephropathy in a Murine Model of Diabetes

**DOI:** 10.3390/ijms25126686

**Published:** 2024-06-18

**Authors:** Cheng-Hsiu Lin, Chun-Ching Shih

**Affiliations:** 1Department of Internal Medicine, Fengyuan Hospital, Ministry of Health and Welfare, Fengyuan District, Taichung City 42055, Taiwan; kenny0139@gmail.com; 2Department of Nursing, College of Nursing, Central Taiwan University of Science and Technology, No. 666 Buzih Road, Beitun District, Taichung City 406053, Taiwan

**Keywords:** *Phyllanthus emblica* L., db/db mice, diabetic nephropathy

## Abstract

Oil-Gan is the fruit of the genus *Phyllanthus emblica* L. The fruits have excellent effects on health care and development values. There are many methods for the management of diabetic nephropathy (DN). However, there is a lack of effective drugs for treating DN throughout the disease course. The primary aim of this study was to examine the protective effects (including analyses of urine and blood, and inflammatory cytokine levels) and mechanisms of the ethyl acetate extract of *P. emblica* (EPE) on db/db mice, an animal model of diabetic nephropathy; the secondary aim was to examine the expression levels of p- protein kinase Cα (PKCα)/t-PKCα in the kidney and its downregulation of vascular endothelial growth factor (VEGF) and fibrosis gene transforming growth factor-β1 (TGF-β1) by Western blot analyses. Eight db/m mice were used as the control group. Forty db/db mice were randomly divided into five groups. Treatments included a vehicle, EPE1, EPE2, EPE3 (at doses of 100, 200, or 400 mg/kg EPE), or the comparative drug aminoguanidine for 8 weeks. After 8 weeks of treatment, the administration of EPE to db/db mice effectively controlled hyperglycemia and hyperinsulinemia by markedly lowering blood glucose, insulin, and glycosylated HbA1c levels. The administration of EPE to db/db mice decreased the levels of BUN and creatinine both in blood and urine and reduced urinary albumin excretion and the albumin creatine ratio (UACR) in urine. Moreover, EPE treatment decreased the blood levels of inflammatory cytokines, including kidney injury molecule-1 (KIM-1), C-reactive protein (CRP), and NLR family pyrin domain containing 3 (NLRP3). Our findings showed that EPE not only had antihyperglycemic effects but also improved renal function in db/db mice. A histological examination of the kidney by immunohistochemistry indicated that EPE can improve kidney function by ameliorating glomerular morphological damage following glomerular injury; alleviating proteinuria by upregulating the expression of nephrin, a biomarker of early glomerular damage; and inhibiting glomerular expansion and tubular fibrosis. Moreover, the administration of EPE to db/db mice increased the expression levels of p- PKCα/t-PKCα but decreased the expression levels of VEGF and renal fibrosis biomarkers (TGF-β1, collagen IV, p-Smad2, p-Smad3, and Smad4), as shown by Western blot analyses. These results implied that EPE as a supplement has a protective effect against renal dysfunction through the amelioration of insulin resistance as well as the suppression of nephritis and fibrosis in a DN model.

## 1. Introduction

Diabetic nephropathy (DN) is a serious and common complication of diabetes mellitus. Diabetic nephropathy is a morbid microvascular complication associated with diabetes [1]. Diabetic nephropathy is the leading cause of end-stage kidney disease worldwide [2] and is an independent risk factor for all-cause and cardiovascular mortality in diabetic patients [3]. Approximately 30–40% of patients with type 2 diabetes develop diabetic nephropathy [4]. Diabetic nephropathy is the main complication of type 2 diabetes and leads to glomerular membrane thickening, mesangial expansion, glomerular hypertrophy, and overt renal disease [5,6,7].

The markers of renal function include urinary albumin excretion, plasma creatinine, urine creatinine, the urine albumin creatine ratio (UACR; the urine P/C ratio), and histological changes within the kidney structures. Blood urea nitrogen (BUN) is a protein metabolite product. Creatinine is a muscle metabolite product. High BUN and creatinine levels indicate renal dysfunction. Urinary albumin excretion indicates impaired renal function [8,9]. Persistent albuminuria is a hallmark of DN.

The postulated functional role of the classical protein kinase C (PKC) isoforms and the signal pathway via vascular endothelial growth factor (VEGF), nephrin, and then proteinuria plays a core role in the development of diabetic nephropathy [10,11,12,13,14]. Protein kinase C (PKC) activation can directly increase the permeability of albumin and other macromolecules through barriers formed by endothelial cells [10,11]. Vascular endothelial growth factor (VEGF) participates in the regulation of glomerular permeability, glomerular endothelial cell growth, and urinary albumin excretion [12,13,14].

Evidence has emphasized the critical role of inflammation in the pathogenesis of DN [15]. The pathogenesis mechanisms of DN are very complicated and remain unclear. In diabetes mellitus, hyperglycemia induces the activation of various signaling pathways involved in diabetic vascular complications through the production of inflammatory cytokines. Renal inflammation and subsequent fibrosis are critical processes leading to end-stage DN [16,17].

In DN, the accumulation of extracellular matrix (ECM) components in the glomerular mesangium and tubulointerstitium causes early glomerular hypertrophy and eventually glomerulosclerosis and tubulointerstitial fibrosis [18]. Collagen IV accumulation is a crucial phenomenon underlying mesangial expansion [19]. Transforming growth factor-β (TGF-β) is a key regulator of fibrosis that promotes the accumulation of ECM. TGF-β induces phosphorylation and activation of Smad signaling pathway components [20]. TGF-β1 is another important factor in the pathogenesis of DN and mediates the inflammatory response, which exacerbates ECM secretion involving fibronectin and collagen accumulation and accelerates glomerulosclerosis in diabetes [21].

db/db mice serve as an animal model of obesity-related diabetes and can be used to study kidney changes due to diabetes [22]. db/db mice are overweight; hyperglycemic and hyperinsulinemic; and exhibit increased kidney weight, glomerular mesangial matrix, and albumin excretion [23]. The concentration of collagen IV increases with DN progression in patients and db/db mice [24].

*P. emblica* L. (Figure 1) is a plant that is widely distributed in tropical and subtropical regions such as Taiwan. The fruit of *P. emblica* possesses various pharmacological activities. Recently, our findings have shown that *P. emblica* extract displays antihyperglycemic activity both in NOD with spontaneous and cyclophosphamide-accelerated Type 1 diabetic mice [25] and in streptozotocin-induced Type 1 diabetic mice [26]. *Triphala*, a well-recognized and highly efficacious polyherbal Ayurvedic medicine consisting of fruits from the plant species *Emblica officinalis* (*Amalaki*), *Terminalia bellerica* (*Bibhitaki*)*,* and *Terminalia chebula* (*Haritaki*), is a cornerstone of gastrointestinal and rejuvenative treatment [27]. Recently, *Triphala* has been demonstrated to ameliorate nephropathy via inhibition of TGF-β1 and oxidative stress in diabetic rats [28]. Nevertheless, *Triphala* consists of fruits of the three plant species, but it is not merely *P. emblica* L. extract that exhibits an ameliorative effect on diabetic nephropathy. More recently, *Emblica officinalis* fruit extract is reported to display nephroprotective effects against malachite green toxicity by restoration of the renal cytoarchitecture and enhanced activity of antioxidant enzymes in a piscine model [29], and these findings are from an in vitro study. However, the ameliorative activity of ethyl acetate extract of *P. emblica* (EPE) is not well defined in db/db mice. Therefore, the present study was designed to clarify EPE’s effects on Type 2 diabetes and DN in an animal model of db/db mouse.

At present, the major clinical treatment method for DN is to control glycemia and improve renal function and reduce both the incidence of end-stage of DN and proteinuria to provide patients with good control. The present study consists of three parts: to investigate whether EPE can regulate target genes expressions including TGF-β1 and collagen IV in vitro or not, to investigate the protective activity and mechanism of EPE on DN by a db/db mouse model, and to clarify the responsible components for the amelioration of DN. We examined the target gene expressions ameliorating DN of EPE by Western blotting test in vitro, and then, HPLC analysis of refractions of EPE were performed.

The animal study was designed to evaluate whether EPE could improve insulin resistance as well as ameliorate DN by the improvement in renal dysfunction (such as albumin excretion) and ameliorative glomerular morphological damage by immunohistochemistry, and to examine the expression levels of targeted genes (including the expression levels of p-PKCα/t-PKCα and VEGF and renal fibrosis biomarkers including TGF-β1, collagen IV, p-Sma and mad protein (Smad)2, p-Smad3, and Smad4) by Western blotting. Finally, this study examined the effects of seven refractions of EPE on the target expression levels of Collagen IV, TGF-β1, VEGF, and inflammatory cytokine kidney injury molecule (KIM)-1 inhuman renal mesangial (HRM) cells to clarify the main ingredients responsible for the protective effects of EPE against diabetic nephropathy from the early stage to the end stage including inflammation, fibrosis, and angiogenesis.

## 2. Results

### 2.1. Effects of EPE on TGF-β1 and Collagen IV In Vitro

To clarify the protective activity of EPE against diabetic nephropathy, we first aimed to determine the anti-fibrotic activity of EPE within the kidney by high-glucose (HG)-induced diabetic status in human renal mesangial cells via an experimental approach. As shown in Figure 2A–C, in human renal mesangial cells, HG induced higher expression levels of TGF-β1 and collagen IV than that of the control group (*p* < 0.001, *p* < 0.001, respectively). In HG-induced condition, treatment with 5 μg/mL EPE and 10 μg/mL EPE significantly decreased the expression levels of TGF-β1 compared with the HG group (*p* < 0.05, *p* < 0.001, respectively) (Figure 2A,B). In HG-induced condition, treatment with 5 μg/mL EPE and 10 μg/mL EPE significantly reduced the expression levels of collagen IV compared with the HG group (*p* < 0.001, *p* < 0.001, respectively) (Figure 2A,C). The results of the experiment showed EPE treatment decreased fibrosis by regulation of TGF-β1 and collagen IV expressions in human renal mesangial cells.

### 2.2. Animal Treatments

#### 2.2.1. Body Weights and Relative Organ Weights

As shown in Figure 3A, the db/db mice (vehicle-treated db/db) had greater initial body weights than did the db/m mice (*p* < 0.001). Following treatment, the db/db mice (vehicle-treated db/db) had greater final body weights than did the db/m mice (*p* < 0.001). The results of the experiment showed the EPE2- and EPE3-treated db/db groups had lower final body weights than did the db/db group (*p* < 0.05 and *p* < 0.01, respectively) (Figure 3A). As shown in Table 1, the relative weights of liver tissue, kidney tissue, and skeletal muscle in the db/db group were lower than those in the db/m group (*p* < 0.001, *p* < 0.001, and *p* < 0.001, respectively). As shown in Table 1, AG-treated db/db mice had reduced relative weights of liver and pancreas tissue compared with those of the db/db group (*p* < 0.05 and *p* < 0.05, respectively). The results of the experiment showed there was no difference in the relative weights of the liver tissue, kidney tissue, pancreas, and skeletal muscle between the EPE1-, EPE2-, and EPE3-treated db/db mice and the db/db group (Table 1).

#### 2.2.2. Blood Glucose, HbA1_C_, and Insulin Levels

To clarify the effect of EPE on the Type 2 diabetes mellitus, blood levels of glucose, HbA1c, and insulin were examined. At the end of the experiment, the db/db mice displayed increased blood levels of glucose, HbA1_C_, and insulin compared with those in the db/m group (*p* < 0.001, *p* < 0.001, and *p* < 0.001, respectively) (Figure 3B–D). EPE1-, EPE2-, and EPE3-treated db/db mice displayed a decrease in the blood levels of glucose (at 7 and 8 weeks of treatment) (Figure 3B), HbA1_C_, and insulin compared with those in the db/db group (Figure 3C,D). These data indicate that EPE treatment lower hyperglycemia and hyperinsulinemia, and reduced HbA1c, implying that EPE treatment could improve insulin resistance.

#### 2.2.3. Renal Function

At the end of the experiment, the db/db mice displayed increased blood levels of BUN and creatinine compared with the db/m group (*p* < 0.001 and *p* < 0.001, respectively). EPE1-, EPE2-, EPE3-, and AG-treated db/db mice showed lower blood levels of BUN and creatinine than the db/db group (Figure 3E,F). At the end of the experiment, the db/db mice showed increased urine volume, urine albumin, urine creatinine, and urine albumin creatinine ratio (UACR) compared with the db/m group (*p* < 0.001, *p* < 0.001, *p* < 0.001, and *p* < 0.001, respectively) (Figure 3G–J). EPE1-, EPE2-, EPE3-, and AG-treated db/db mice showed lower levels of urine albumin and urine creatinine than the db/db group (Figure 3H,I). EPE2-, EPE3-, and AG-treated db/db mice exhibited a decrease in the levels of urine volume and UACR compared with levels in the db/db group (Figure 3G,J). At the end of the experiment, the db/db mice displayed increased urine levels of sodium ions, potassium ions, and chloride iron compared with those of the db/m group (*p* < 0.001, *p* < 0.001, and *p* < 0.001, respectively) (Figure 3K–M). EPE3- and AG-treated db/db mice had decreased urine sodium ion and chloride ion levels compared with those in the db/db group (Figure 3K,M). EPE1-, EPE2-, EPE3-, and AG-treated db/db mice had significantly lower urine potassium ion levels than the db/db group (*p* < 0.001, *p* < 0.001, *p* < 0.001, and *p* < 0.001, respectively) (Figure 3L).

#### 2.2.4. Blood Levels of Inflammatory Cytokines

To clarify the effect of EPE on the inflammation-induced nephritis, the various inflammatory cytokines kits were used. The db/db mice displayed increased blood levels of kidney injury molecule-1 (KIM-1), C-reactive protein (CRP), NLR family pyrin domain containing 3 (NLRP3), IL-18, and TNF-α compared with those of the db/m group (*p* < 0.001, *p* < 0.001, *p* < 0.001, *p* < 0.001, and *p* < 0.001, respectively) (Figure 3N–R). EPE1-, EPE2-, EPE3-, and AG-treated db/db mice showed reduced blood levels of KIM-1, CRP, and NLRP3 compared with the levels in the db/db group (Figure 3N–P). EPE2-, EPE3-, and AG-treated db/db mice showed decreased blood levels of IL-18 and TNF-α compared with the levels in the db/db group (Figure 3Q,R). These data indicate that EPE treatment reduced kidney injury and inflammation.

#### 2.2.5. Morphological and Immunohistochemical Analyses of the Kidney

To clarify the effect of EPE on the kidney glomerulus, the glomeruli were stained with periodic acid Schiff (PAS) stain. Histological PAS staining of the glomerular basement membrane revealed glomerular basement membrane thickening and mesangial expansion, as well as increased accumulation of ECM in the db/db group compared with those in the db/m group. EPE1-, EPE2-, EPE3-, and AG-treated db/db mice showed not only reduced thickness of the glomerular basement membrane but also less mesangial matrix expansion compared with the db/m group (Figure 4A,B). IHC staining revealed that EPE2- and EPE3-treated db/db mice had increased nephrin expression levels and Nephrin-positive area (%) score by IHC stains compared with those of the db/db group (Figure 4C,D).

#### 2.2.6. Target Gene Expression Levels by Western Blotting Analysis

As shown in Figure 5A,B, db/db mice showed decreased expression levels of p-PKCα/t-PKCα and nephrin, but increased expression levels of VEGF in the kidney compared with those of the db/m group (*p* < 0.001, *p* < 0.001, and *p* < 0.001, respectively). EPE2-, EPE3-, and AG-treated db/db mice showed increased expression levels of p-PKCα/t-PKCα but reduced expression levels of VEGF in the kidney compared with the levels in the db/db group. EPE1-, EPE2-, EPE3-, and AG-treated db/db mice had increased expression levels of nephrin in the kidney compared with those in the db/db group (*p* < 0.001, *p* < 0.001, *p* < 0.001, and *p* < 0.001, respectively) (Figure 5A,B). The db/db mice showed increased expression levels of TGF-β1, collagen IV, and fibronectin in the kidney compared with those of the db/m group (*p* < 0.001, *p* < 0.001, and *p* < 0.001, respectively) (Figure 5C,D). EPE2-, EPE3-, and AG-treated db/db mice had decreased expression levels of TGF-β1, collagen IV, and fibronectin in the kidney compared with the levels in the db/db group (Figure 5C,D). The db/db mice showed increased expression levels of p-Smad2/t-Smad2, and p-Smad3/t-Smad3 in the kidney compared with those of the db/m group (*p* < 0.001, *p* < 0.001, *p* < 0.001, respectively) (Figure 5E,F). EPE2-, EPE3-, and AG-treated db/db mice had decreased expression levels of p-Smad2/t-Smad2 and p-Smad3/t-Smad3 in the kidney compared with those in the db/db group (Figure 5E,F). The db/db mice showed increased expression levels of p-NLRP3/t-NLRP3, ICAM-1, and capalape-1 in the kidney compared with those of the db/m group (*p* < 0.001, *p* < 0.001, and *p* < 0.001, respectively) (Figure 5G,H). EPE2-, EPE3-, and AG-treated db/db mice had decreased expression levels of p-NLRP3/t-NLRP3 and ICAM-1, and EPE3-treated db/db mice had reduced expression levels of capalase-1 in the kidney compared with those in the db/db group (Figure 5G,H).

### 2.3. Effects of the Seven Fractions of EPE on Targeted Gene Expression In Vitro

Following the animal study, we explored the main active ingredient of EPE and the expression levels of collagen IV, TGF-β1, VEGF, and KIM in human renal mesangial (HRM) cells, which are responsible for its fibrogenesis, angiogenesis, and inflammatory effects, respectively. The preparation of seven fractions of EPE was described in a previous paper [25]. The fruits of *P. emblica* (126.3 g) were extracted with methanol at 25 °C (3 × 7 d). The methanol extract was dried under vacuum, and the remaining sample (31.4 g) was obtained. The remaining sample was suspended in H_2_O and partitioned with EtOAc. The EtOAc-soluble fraction (5.6 g) was subjected to column chromatography on silica gel with a gradient of *n*-hexane and EtOAc as the mobile phase, and seven fractions were obtained [25]. The seven fractions of EPE (EA) are described as follows: 2~10%, EA fraction 1 (EA-1); 10~20%, EA fraction 2 (EA-2); 20~30%, EA fraction 3 (EA-3); 30~50%, EA fraction 4 (EA-4); 50~70%, EA fraction 5 (EA-5); 70~100%, EA fraction 6 (EA-6); and 100%, EA fraction 7 (EA-7). Figure 6A–E show that the glucose-treated groups had increased expression levels of Collagen IV, TGF-β1, VEGF, and KIM in comparison to the control group (*p* < 0.001, *p* < 0.001, *p* < 0.001, and *p* < 0.001, respectively). EA-1, EA-4, EA-5, EA-6, and EA-7 significantly decreased the expression levels of collagen IV (Figure 6A,B). EA-3, EA-4, EA-5, EA-6, and EA-7 reduced the expression levels of TGF-β1 (Figure 6A,C). EA-4, EA-6, and EA-7 decreased the expression levels of KIM (Figure 6A,D).

### 2.4. Analysis of the EA-4 Fraction of EPE

Polyphenolic compounds were found in the EPE fraction in our recent publication [25,26]. Our previous study showed that the phenolic compounds found in EPE were gallic acid (3.28%), chebulagic acid (6.44%), and ellagic acid (2.23%) [26].

In this study, the polyphenolic compounds in the EA-4 sample were evaluated. The retention time for chebulagic acid, which was used as a reference compound to qualitatively and quantitatively analyze the EA-4 sample, was 14.447 min, as shown in Figure 6F,G. The major absorption peaks of phenolic acids were consistent with the photodiode array (PDA) spectra of the reference compound.

As shown in Figure 6F,G and Appendix A, the main ingredient in EA-4 is the phenolic compound in chebulagic acid (54.4%).

## 3. Discussion

The fruits of the genus *P. emblica* L. have excellent effects on health care. Although there are many methods for the treatment of DN, there is a lack of effective drugs for treating DN throughout the disease course. The present study was designed to evaluate the protective effects of EPE on DN, to determine whether EPE can ameliorate renal dysfunction (including urine analyses, the measurement of inflammatory cytokine levels, and histological examination) and ameliorate fibrosis within the kidney, and to clarify the underlying molecular mechanisms of action of EPE; the secondary aim was to examine the expression levels of PKCα in the kidney and its downregulation of VEGF and fibrosis genes TGF-β1 and Smads by Western blot analyses. The present study demonstrated that EPE treatment not only regulated urine volume but also lowered blood glucose, HbA1c, and insulin levels, suggesting that EPE could improve insulin resistance by modulating blood glucose and insulin levels in individuals with type 2 diabetes. Our findings showed that in db/db mice, overweight, hyperglycemia and hyperinsulinemia were induced, and plasma creatinine, urine creatinine, and the urine albumin creatine ratio (urine P/C ratio) and histological changes in the glomerular mesangial matrix were consistent with the findings of a previous study [23]. Moreover, the present study showed that db/db mice displayed higher plasma creatinine levels, urine creatinine levels, and urine albumin creatine ratios (urine P/C ratios) than db/m mice. Urinary albumin excretion may be a predictive factor for the prognosis of DN, and our studies demonstrated that EPE treatment decreased urinary albumin displayed a kidney-protective effect in type 2 diabetes mellitus. Since the postulated functional role of the classical PKC isoform activation and the signal pathway including VEGF, nephrin, and proteinuria plays a role in the development of diabetic nephropathy [10,11,12,13,14], the present study was designed to clarify whether EPE ameliorates diabetic nephropathy and reduces proteinuria by PKC isoform activation and regulation of target genes including VEGF and nephrin expressions.

Activation of the diacylglycerol (DAG)-protein kinase C (PKC) pathway, enhancement of the polyol pathway, and overproduction of advanced glycation end products have all been proposed as potential cellular mechanisms by which hyperglycemia induces diabetic vascular complications [30]. PKC activation is also known to mediate several biological actions of VEGF, including its ability to increase capillary permeability [31]. Our findings showed that administration of EPE2 and EPE3 to db/db mice increased the expression levels of p-PKCα/t-PKCα but decreased the expression levels of VEGF in the kidney, thus contributing to a decrease in urinary albumin excretion.

DN is characterized by glomerular alterations in renal tissue, including glomerular basement membrane thickening and mesangial matrix expansion, leading to the development of glomerulosclerosis [32]. Increased mesangial cell proliferation and the accumulation of ECM components such as collagen in the glomeruli are characteristic pathologic features of early-stage DN [33]. These histological characteristics were detected in the PAS-positive mesangial matrix area of db/db mice. According to previous studies [30,31,32,33], our histological results showed that the administration of EPE to db/db mice decreased kidney glomerulus mesangial matrix expansion, and moreover, EPE reduced the expression levels of VEGF, TGF-β1, and collagen IV, suggesting that EPE could ameliorate kidney dysfunction and decrease urinary albumin in a mouse model.

Persistent albuminuria is a hallmark of DN that occurs with damage to glomerular podocytes. Podocytes make up the slit diaphragms that function as the final barrier against the flow of macromolecules into the urine [34]. It has been suggested that the initial stages of the loss of the permeability barrier in DN are associated with nephrin [6]. Nephrin, a 180 kDa transmembrane protein in podocytes, was the first molecule identified in the slit between podocyte foot processes and is a major component of the slit diaphragm [35]. Reduced nephrin levels may also lead to the development of proteinuria. Nevertheless, nephrin should be expressed in most glomeruli, and some previous studies have shown no noticeable manifestations within glomeruli [36,37]. The depletion of nephrin and podocin proteins in podocytes following glomerular injury causes severe proteinuria [38]. In the present study, histological analysis revealed that the administration of EPE to db/db mice increased the expression of nephrin and reversed the increase in the expression of nephrin in glomerular injury. This finding implies that EPE could effectively ameliorate renal dysfunction damage and albuminuria and that EPE may be a therapeutic option for preserving nephrin to slow nephropathy progression. Our findings showed that EPE could ameliorate the incidence of diabetic nephropathy, decrease renal inflammatory cytokines, and reverse the expression of nephrin within the kidney following kidney damage-induced proteinuria in db/db mice.

The results presented in Table 1 suggest that the AG-treated db/db mice had reduced relative weights of liver and pancreas tissue compared with those of the db/db group. The results of the experiment showed that there were no differences in the relative weights of liver tissue, kidney tissue, pancreas, and skeletal muscle between db/db mice treated with EPE1, EPE2, and EPE3 and the db/db group. db/db mice normally have reduced relative weights compared to controls, and EPE1, EPE2, and EPE3 treatments for diabetes and diabetic nephropathy were expected to restore the normal weight of these organs, but this is not reflected in the present results. An explanation for the abnormality of EPE treatments is the regulation of PCKα and downregulation of expression levels of VEGF, thus contributing to a decrease in proteinuria and, in turn, a reduction in kidney hypertrophy to improve renal function and finally reducing the progress rate of DN; thus, there is no difference in relative weights of kidney tissue between EPE1, EPE2, and EPE3 treatments and db/db mice. Another explanation for the abnormality may be the complex physiological phenomena among peripheral tissues in murine body not easily interpreted.

Inflammatory cytokines play a core role in the development and progression of DN [15]. NLR family pyrin domain containing 3 (NLRP3) undergoes oligomerization in the presence of the adaptor protein apoptosis-associated speck-like protein (ASC) and protease caspase-1 to form a protein complex termed the inflammasome. Inflammasome formation is important for the autoprocessing of caspase-1 and the activation of the cytokines pro-IL-1b and pro-IL-18 [39]. Mononuclear cell invasion and abnormal expression of inflammatory mediators, including intercellular adhesion molecule-1 (ICAM-1) and TGF-β1, are observed in renal tissues at early stages of DN [40]. Administration of EPE to db/db mice decreased blood levels of proinflammatory cytokines (including KIM-1, CRP, NLRP3, IL-1, and IL-18) and reduced the expression levels of p-NLRP3/t-NLRP3, ICAM-1, caspase-1, and TGF-β1 in the kidney. Renal inflammation and subsequent fibrosis are critical processes leading to end-stage DN [15,16]. Thus, EPE ameliorates renal inflammation through the inhibition of inflammatory cytokines in diabetic db/db mice, suggesting that EPE protects against DN-induced fibrosis and inflammation in type 2 diabetic db/db mice.

DN is a morbid microvascular complication associated with diabetes and is the most common cause of end-stage renal disease [1]. In DN, the accumulation of ECM components in the glomerular mesangium and tubulointerstitium causes early glomerular hypertrophy and eventually glomerulosclerosis and tubulointerstitial fibrosis [18]. TGF-β1 is another important factor in the pathogenesis of DN and mediates the inflammatory response, which exacerbates ECM secretion involving fibronectin and collagen accumulation and accelerates glomerulosclerosis in diabetes [21]. A previous study suggested that TGF-β1 mediates ECM accumulation in mesangial and tubular cells and that the suppression of TGF-β1 signaling significantly reduces renal fibrosis and decreases the mRNA levels of major mediators of ECM deposition in db/db mice [41]. TGF-β functions as a key regulator of fibrosis by promoting the accumulation of ECM. TGF-β induces the phosphorylation and activation of Smad signaling pathway components [20]. In the present study, the administration of EPE to db/db mice decreased the expression levels of VEGF and TGF-β1 in the kidney, as determined by Western blotting. The concentration of collagen IV increases with DN progression in patients and db/db mice [24]. Collagen IV accumulation is a crucial phenomenon underlying mesangial expansion [19]. The mesangial expansion and glomerular fibrosis observed in db/db mice may result from molecular changes within the renal tissue, including the activation of various proinflammatory cytokines and growth factors [42]. In the present study, we used aminoguanidine (AG) as a comparative drug since aminoguanidine (AG) is an investigational drug for the management of diabetic nephropathy, although it has side effects [43]. In the present study, histological PAS staining of the glomerular basement membrane revealed glomerular basement membrane thickening and mesangial expansion, as well as increased ECM accumulation, in the db/db group compared with the db/m group. Compared with those in the db/m group, the glomerular basement membrane thickness in the EPE- and AG-treated db/db groups was reduced, and mesangial matrix expansion was ameliorated. Moreover, EPE treatment decreased the renal expression levels of p-Smad-2 and p-Smad-3, reduced the expression levels of fibronectin and collagen IV, and decreased TGF-β1/Smad activation, which were associated with the inhibition of ECM accumulation and tubulointerstitial fibrosis, thus contributing to the suppression of severe renal fibrosis and a decrease in the cumulative incidence of diabetic nephropathy in 20-week-old db/db mice.

Following the animal study, we attempted to identify the main constituent of EPE by fractioning EPE into seven fractions, and our previous in vitro findings showed that EA-6 has the greatest effect on Akt phosphorylation and membrane GLUT4 expression, which play a core role in diabetic target gene pathogenesis [25]. In this previous HPLC analysis, the study showed that the EA-6 fraction of EPE yielded > 33.4% of the total polyphenol gallic acid extracted and that this fraction represented the main component responsible for the antidiabetic activity of EPE in NOD with spontaneous and cyclophosphamide-accelerated Type 1 diabetic mice [25]. The phenolic compounds found in EPE were gallic acid (3.28%), chebulagic acid (6.44%), and ellagic acid (2.23%) in streptozotocin-induced Type 1 diabetic mice [26].

To clarify the main components responsible for the amelioration of DN status of EPE, we analyzed the expression levels of collagen IV, TGF-β1, VEGF, and KIM in HRM cells, which are responsible for its fibrotic, angiogenic, and inflammatory effects within the kidney, respectively. Our previous studies [25,26] showed that the main ingredient of EA-6 was gallic acid. The present HPLC analysis showed the main ingredient of EA-4 was chebulagic acid (Figure 6F,G and Appendix A). Chebulagic acid accounted for >54.4% of the total polyphenolic compounds extracted. Chebulagic acid has an antiangiogenic effect that is associated with VEGFR2 signaling pathway inhibition [44]. In the present study, our findings showed that EA-4 and EA-6 displayed the greatest decreases in expression levels of collagen IV and TGF-β1 for anti-fibrotic activity in vitro, and EA-4 and EA-6 displayed the greatest decreases in expression levels of VEGF for anti-angiogenic activity, and EA-4 and EA-6 displayed the greatest decreases in expression levels of KIM for renal anti-inflammatory activity. An explanation for EPE’s protective activity from diabetic nephropathy is that the major constituents of the fruit extract include the total polyphenol components gallic acid and chebulagic acid, which display the majority of anti-inflammatory, anti-fibrotic, and anti-angiogenic activity, and the indirect modulation of inflammatory cytokines and mutual restraint and suppression of nephritis, fibrosis, and proteinuria, thus contributing to the preventive effects of EPE on DN status for the early stage to the end stage.

## 4. Materials and Methods

### 4.1. Chemicals

Antibodies against Smad-2 (no. GTX111075), Smad-3 (no. GTX34208), Smad-4 (no. GTX112980), phospho-Smad2 (Ser250) [GT1291] (no. GTX03203), and phospho-Smad3 (Ser423/425) (no. GTX129841) were obtained from GeneTex International Corporation (East Dist., Hsinchu City, Taiwan). Mouse VEGF164 antibody (AF-493-NA) (antigen affinity-purified polyclonal goat IgG) was obtained from R&D Systems Biotechnology (Minneapolis, MN, USA). Antibodies against nephrin (sc-32530), ICAM-1/CD54 (sc-8439), MCP-1 (sc-32771), ASC (sc-271054), and caspase-1 (sc-56036) were obtained from Santa Cruz Biotechnology, Inc. (Dallas, TX, USA). Mouse VEGF164 antibody (AF-493-NA) (antigen affinity-purified polyclonal goat IgG) was obtained from R&D Systems Biotechnology (Minneapolis, MN, USA). An antibody against TGF-β1 (MA5-15065) was obtained from Invitrogen, Inc. (Carlsbad, CA, USA). A mouse NLRP3/NALP3 monoclonal antibody (MBS6002814) and an NLRP3 polyclonal antibody (phospho-Ser295) (MBS9430199) were purchased from MyBioSource, Inc. (San Diego, CA, USA). Antibodies against β-actin (8H10D10) (no. 3700), phospho-PKCα/β II (Thr638/641) (no. 9375), and PKCα (no. 2056) were obtained from Cell Signaling Technology Inc. (Beverly, MA, USA). Horseradish peroxidase (HRP)-conjugated goat anti-mouse IgG (H+L)-AP (TAAB01) was obtained from BIOTnA (Kaohsiung, Taiwan). 3,3′-diaminobenzidine tetrahydrochloride (D7304-1SET, DAB) was obtained from Millipore (Bedford, MA, USA).

### 4.2. Preparation of the Ethyl Acetate Extract of P. emblica L. (EPE)

Fruits of *P. emblica* L. (Figure 1) were purchased from Taiwan Miaoli Amla Cooperative (Houlong Township, Miaoli County, Taiwan). These fruits were identified by the Department of Chinese Pharmaceutical Sciences and Chinese Medicine Resources of China Medical University, Taiwan, where the voucher specimen (CMPF385) was deposited. Fruits of *P. emblica* L. (Figure 1) were purchased from Taiwan Miaoli County, Ogan Marketing Cooperation. A total of 7.3 kg of fruits of *P. emblica* L. were dried and pulverized into 1.34 kg of fine powder. This procedure was performed as described in our previous study [25,26]. The fine powder was extracted with methanol three times and concentrated, and then, the crude methanolic extract was subjected to suspension in H_2_O and partitioning with EtOAc, and then, the H_2_O fraction and the EtOAc fraction were obtained. The EtOAc fraction was used for the animal study [25,26].

### 4.3. Cell Culture and Assessment of the Expression Levels of TGF-β1 and Collagen IV in Human Renal Mesangial (HRM) Cells

Human renal mesangial cell (HRMC) (no. 4200; ScienCell Research Laboratories (Carlsbad, CA, USA) lysates were treated with 1 μg/mL, 5 μg/mL, or 10 μg/mL EPE for 24 h, and then, the cells were collected to examine the expression levels of TGF-β1 and collagen IV. The culture procedure consisted of culturing HRM cell lysates in DMEM containing 25 mmol/L glucose and 10% FBS, which resembles the diabetic hyperglycemic environmental state. Human renal mesangial cells treated with EPE in high-glucose conditions were subjected to Western blot analyses with primary antibodies against TGF-β1 and collagen IV. The protein concentration was analyzed via the BCA assay (Pierce), and equal amounts of protein were then diluted with 4× diluted SDS sample buffer and subjected to SDS–PAGE. Density blotting was performed as described in previous reports [45,46].

### 4.4. Animal Treatment

Forty male db/db mice (BKS. Cg-m+/+Lepr^db^NJU) and eight male age-matched nondiabetic db/m (C57BL/6J Lepr) mice were purchased from the National Laboratory Animal Breeding Center at 11 weeks of age. All the mice were handled under the guidepost of the school and according to the guidelines of the Animal Ethics Committee (approval no. 110-CTUST-014), and the welfare-related assessments ensued, and a particular small sample size in each group was determined. All the mice were randomly divided into 6 groups (*n* = 8 per group; total number = 48) as follows. In Group I, db/m mice were treated with vehicle. After one week of acclimation, forty db/db mice were randomly divided into the following 5 groups: group II, the db/db control group, in which diabetic mice were given the same volume of distilled water orally; groups III, IV, V, and VI, in which diabetic mice were orally given 100, 200, or 400 mg/kg EPE (which are referred to as EPE1, EPE2, and EPE3, respectively); and group VI, in which diabetic mice were orally given aminoguanidine (AG) (20 mg/kg). Treatments were given orally once daily in the morning for 8 weeks. After 10 h of fasting, blood was obtained from the retro-orbital sinuses of the mice for the assessment of glucose concentrations. All procedures (such as the collection of peripheral organs and analysis of biochemical parameters and cytokines) were performed as described in previous studies [45,46,47,48]. Twenty-four hours before the end of the experiment, urine samples were collected, and the urine volume was recorded.

#### 4.4.1. Collection of Plasma, Urine, and Kidney Samples

After 8 weeks of treatment, blood and kidney samples were collected. The kidneys and adipose and liver tissues were dissected according to defined anatomical landmarks. The weights of kidney, adipose, and liver tissues were measured. Visceral fat was defined as the sum of epididymal white adipose tissue and retroperitoneal white adipose tissue, and then, all tissue including kidney tissues were immediately stored at −80 °C until use. Then, the plasma was obtained from the coagulated blood by centrifugation at 1600× *g* for 15 min at 4 °C and immediately frozen at −80 °C until use.

#### 4.4.2. Measurements of Blood Glucose, Insulin, and HbA1_C_ Levels

Blood samples were obtained from the retro-orbital sinus of 12 h fasted mice. Blood glucose levels were measured by the glucose oxidase method by a Glucose membrane kit (YSI2365-1, GlaxoSmithKline Pharmaceuticals Ltd., Middlesex, UK), and insulin levels by a mouse Insulin ELISA kit (no. 10-1113-01, Mercodia, Uppsala, Sweden) were measured by commercial enzyme-linked immunosorbent assay (ELISA) kits according to the manufacturer’s instructions as in previous procedures [25,26]. Blood percent HbA1c was measured with a Hemoglobin A_1C_ kit (BioSystems S.A., Barcelona, Spain).

#### 4.4.3. Analysis of Blood Inflammatory Cytokine Levels

Blood samples were obtained from the retro-orbital sinus of 12 h fasted mice. Blood inflammatory cytokine levels were measured by a commercial mouse C-reactive protein (CRP) ELISA kit (LS-F4264, LSBIO, Ltd., Seattle, WA, USA), a commercial mouse KIM-1 ELISA kit (LS-F24859, LSBIO, Ltd., Seattle, WA, USA), and a mouse NALP3/NLRP3 ELISA kit (no. LS-F32087, LSBIO, Ltd.) according to the manufacturer’s instructions.

#### 4.4.4. Determination of Renal Function

The mice were maintained in separate metabolic cages for two days at the end of the experiment for the collection of urine samples for 24 h (from 2:00 P.M. to 2:00 A.M.) for the determination of urine volume, water and food intake, and ion concentrations. Urine samples were adapted to examine the levels of ions (including sodium ions, potassium ions, and chloride ions). A portion of the blood and urine samples was used to assess the BUN concentrations and plasma levels of creatinine (no. 99-0013052) both in urine and blood using a Catalyst One chemistry analyzer (IDEXX Laboratories, Westbrook, ME, USA) and commercial kits according to the manufacturer’s instructions. Urinary albumin excretion was determined by a mouse urinary albumin ELISA kit (No. MBS3806806) from MyBioSource, Inc. (San Diego, CA, USA), and creatinine levels were examined with a mouse creatinine assay kit (MBS763433) from MyBioSource, Inc. (San Diego, CA, USA); then, the urine albumin creatine ratio (UACR; urine P/C ratio) was calculated.

#### 4.4.5. Morphological and Immunohistochemical Staining

For the evaluation of the glomerular basement membrane and mesangial expansion in the kidney, kidney slices were stained with a periodic acid Schiff (PAS) stain kit (TASS01-250; BIOTnA, Kaohsiung, Taiwan) followed by counterstaining with hematoxylin (TA01NB; BIOTnA, Kaohsiung, Taiwan) according to the manufacturer’s instructions.

For immunohistochemical staining, kidney sections were stained with an anti-nephrin antibody (1:500 dilution) [31] at 4 °C overnight. This procedure was performed as described in a previous study [25]. Afterward, the sections were stained with a horseradish peroxidase-conjugated goat anti-mouse IgG (H+L)-AP secondary antibody (1:1000 dilution) at room temperature for 2 h. Immunoreactive areas were detected by 3,3′-diaminobenzidine tetrahydrochloride followed by counterstaining with hematoxylin (TASS17; BIOTnA, Kaohsiung, Taiwan). Digital images were taken with a microscope (OLYMPUS model: BX53 microscope), OLYMPUS model: D28 camera, and OLYMPUS cellScens Image Software for DP23 at 10 (ocular) × 20 (object lens) magnification.

#### 4.4.6. Western Blotting Assay

Kidney tissues were immediately removed and quickly homogenized with RIPA buffer (Sigma, St. Louis, MO, USA) prior to Western blotting. Kidney samples were separated on 4–12% Bis Tris gradient gels (Life Technologies, Carlsbad, CA, USA) and then transferred to a membrane, followed by the addition of the following primary antibodies at 4 °C overnight: an anti-nephrin antibody (1:500 dilution); antibodies against Smad-2, Smad-3, Smad-4, phospho-Smad2 (Ser250), phospho-Smad3, collagen IV, VEGF164, TGF-β1, MCP-1, ASC, caspase-1; NLRP3/NALP3 monoclonal antibody; NLRP3 (phospho-Ser295) polyclonal antibody; CASP1 antibody; and antibodies against β-actin, phospho-PKCα/β II (Thr638/641), and PKCα. Then, the sections were incubated with the following secondary antibody at room temperature for 2 h: HRP-conjugated goat anti-mouse IgG (H+L)-AP (1:1000 dilution; TAAB01, BIOTnA, Kaohsiung, Taiwan). Immunospecific proteins were detected by Enhanced Chemiluminescent (ECL) (EMD Millipore, Germany) HRP Substrate, and then, the PVDF membrane was transferred into the Chemiluminescent electrical gel analysis system (MultiGel-21; TOPBIO), and the images were taken as described in previous studies [45,46,47,48].

### 4.5. Analysis of the Effects of the Seven Fractions of EPE on Targeted Gene Expression In Vitro

Following the animal study, we aimed to explore the main component of EPE responsible for its functional activities. Since collagen IV, TGF-β1, KIM, and VEGF play a critical role in the molecular mechanisms of diabetic nephropathy from the early stage of inflammation to fibrogenesis, and angiogenesis, and thus, cell culture analysis of the expression of KIM (no. ab47635; Abcam (Discovery Drive, Cambridge Biomedical Campus, Cambridge, UK)), collagen IV (no. 75087s; Cell Signaling Technology (Beverly, MA, USA)), TGF-β1 (no. 3709ss; Cell Signaling Technology), and VEGF (no. 2463s; Cell Signaling Technology) was performed using Western blotting with antibodies specific to these proteins to clarify the anti-fibrotic, anti-inflammatory, and anti-angiogenic activity in diabetic nephropathy. Human renal mesangial cells (HRMC) (no. 4200, ScienCell, USA) were cultured in Mesangial Cell Medium (MsCM) (no. 4201, ScienCell, USA) containing 25 mmol/L glucose and 10% FBS. Under normal culture conditions, the cell lysates in the MsCM containing 25 mmol/L glucose and FBS or the individual fractions of 20 μmol/L of ethyl acetate from *P. emblica* (including EA-1~EA-7) were collected, followed by 24 h of stimulating the HRM cells. Then, the cells were collected to analyze the expression levels of collagen IV, TGF-β1, KIM, and VEGF by Western blotting. The seven fractions were prepared as previously described [25]. The fruits of *P. emblica* (126.3 g) were extracted with methanol at 25 °C (3 × 7 d).

### 4.6. Chemical and HPLC Analyses

#### 4.6.1. Fingerprint Analysis by HPLC

The analysis was performed on a HITACHI high-performance liquid chromatographic (HPLC) L-5000 system equipped with a degasser, pumps, and a photodiode array detector linked to a PC computer running the software program HPLC LACHROM as described in our previous paper [25].

#### 4.6.2. Determination of Phenolic Compound Contents

The method was performed according to a previous report [25]. The mobile phase contained acetonitrile (solvent A) and acidified water with trifluoroacetic acid (0.05%, solvent B) as described in a previous paper [25]. This analysis was designed to examine whether polyphenolic compounds (including gallic acid, chebulagic acid, and ellagic acid) were found in the EPE fraction.

### 4.7. Statistical Analysis

The results were analyzed by SPSS software (SPSS Inc., Chicago, IL, USA), and nonparametric analysis was performed via the Kruskal–Wallis H test, followed by the Mann–Whitney *U* test. The results are presented as the means and standard errors. A *p* value less than 0.05 was considered to indicate statistical significance.

## 5. Conclusions

In conclusion (Figure 7), the present study demonstrated that EPE treatment not only regulated urine volume but also lowered blood glucose, HbA1c, and insulin levels, suggesting that EPE could improve insulin resistance by modulating blood glucose and insulin levels in individuals with type 2 diabetes. EPE treatment could improve renal dysfunction and reduce proteinuria by reversal of Nephrin expressions and regulating PKCα activation and VEGF expression and suppressing glomerular basement membrane thickening and mesangial matrix expansion, but also reduce inflammatory cytokines, thus contributing to a decrease in ECM accumulation. The present study clearly demonstrated the protective effects and molecular mechanisms of EPE in a mouse model of DN (db/db mice), and the findings further suggested that chebulagic acid is the main component of EPE responsible for its antiangiogenic effect of VEGF suppression, which contributes to its ability to ameliorate DN. Therefore, chebulagic acid may be a DN treatment candidate throughout the disease course in humans. These results suggest that EPE could improve kidney function. In conclusion, EPE treatment ameliorated diabetic nephropathy in db/db mice.

## Figures and Tables

**Figure 1 ijms-25-06686-f001:**
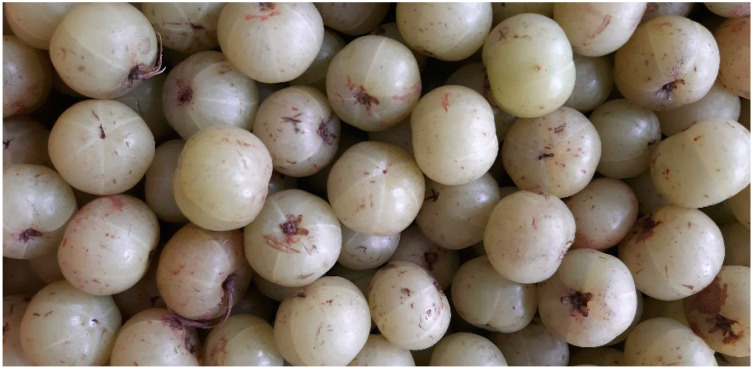
Fruits of *P. emblica* L.

**Figure 2 ijms-25-06686-f002:**
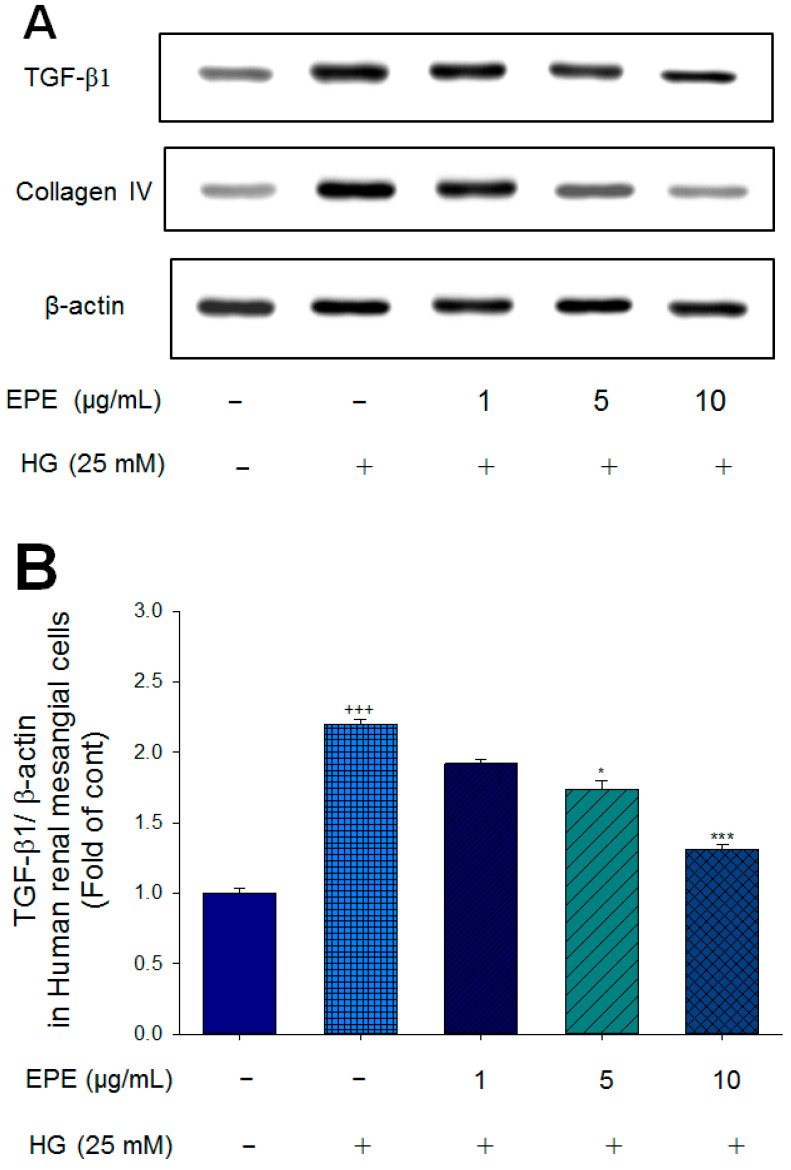
The targeted gene effects of ethyl acetate extract of *P. emblica* L. (EPE) in human renal mesangial cell lysates by Western blot analyses with a primary antibody against TGF-β1 and Collagen IV. (**A**) Representative blots in human renal mesangial cells; (**B**,**C**) Quantification of the expression levels of TGF-β1 and Collagen IV. All values are means ± S.E. +++ *p* < 0.001 compared with the control group; * *p* < 0.05, *** *p* < 0.001 compared with the HG group. The β-actin was used as the internal standard in each sample. HG: high glucose. EPE: ethyl acetate extract of *P. emblica* L.

**Figure 3 ijms-25-06686-f003:**
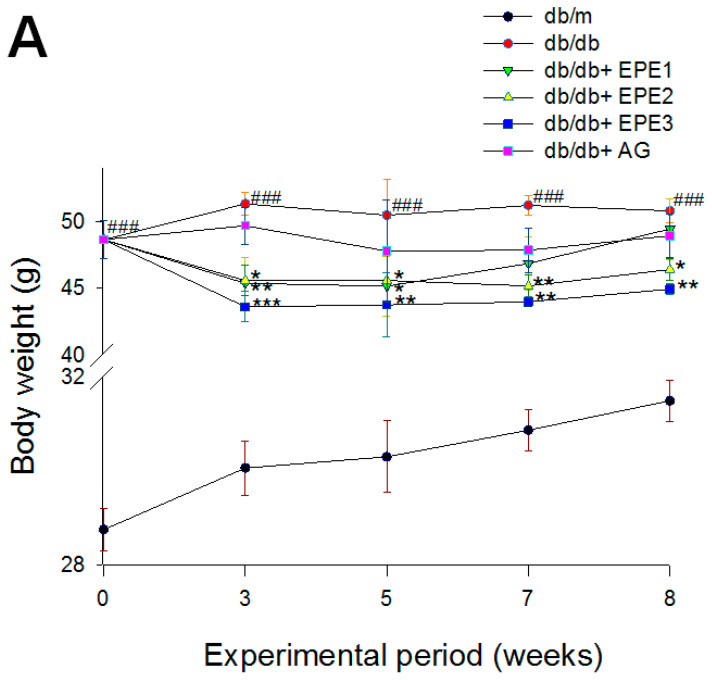
Effects of ethyl acetate extract of *P. emblica* L. (EPE) in db/db mice. ^###^ *p* < 0.001 compared with the db/m group; * *p* < 0.05, ** *p* < 0.01, *** *p* < 0.001 compared with the db/db + vehicle (distilled water) (db/db) group. EPE, ethyl acetate extract of *P. emblica* L. EPE1: 100, EPE2: 200, EPE3: 400 mg/kg body weight; AG: aminoguanidine (20 mg/kg body weight).

**Figure 4 ijms-25-06686-f004:**
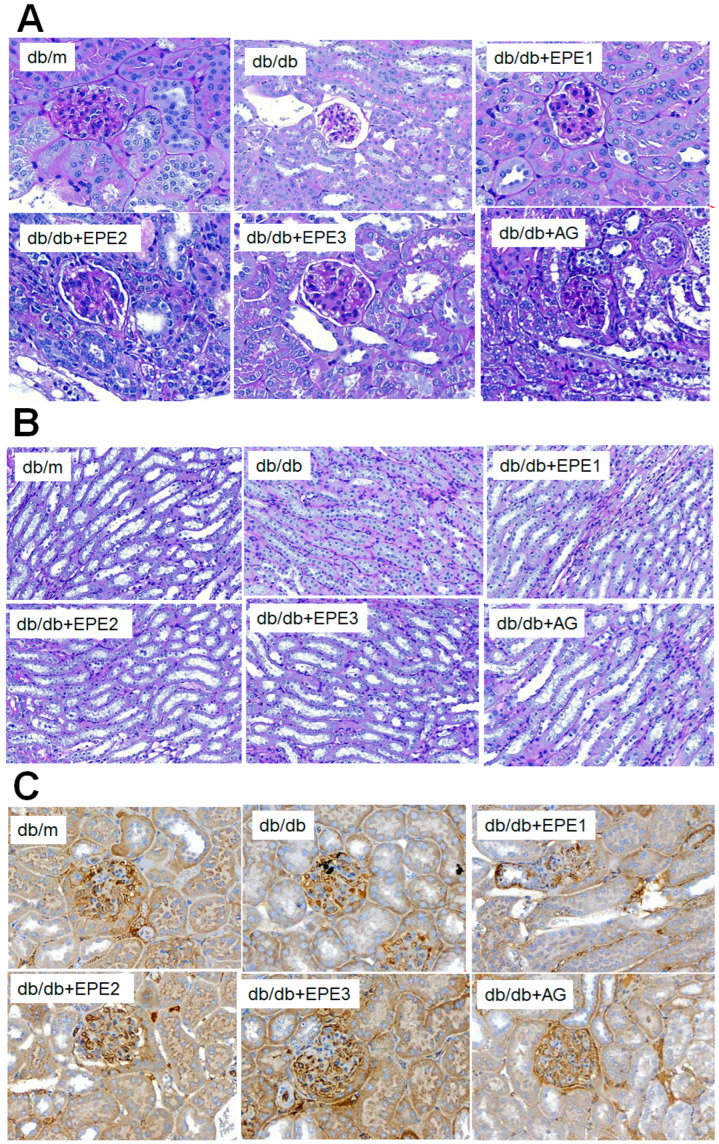
Representative photographs of the kidney; ethyl acetate extract of *P. emblica* L. (EPE) in db/db mice on (**A**) glomerular basement membrane in cortex and (**B**) mesangial matrix expansion in outer medulla by periodic acid Schiff (PAS) staining (magnification: 20 × 10); (**C**) expression levels of nephrin in the kidneys by immunohistochemical (IHC) staining; (**D**) Nephrin-positive area (%) score by IHC stains compared with the db/db group. EPE: EPE1, EPE2, and EPE3 (100, 200, and 400 mg/kg body). ^###^ *p* < 0.001 compared with the db/m group; *** *p* < 0.001 compared with the db/db + vehicle (distilled water) (db/db) group.

**Figure 5 ijms-25-06686-f005:**
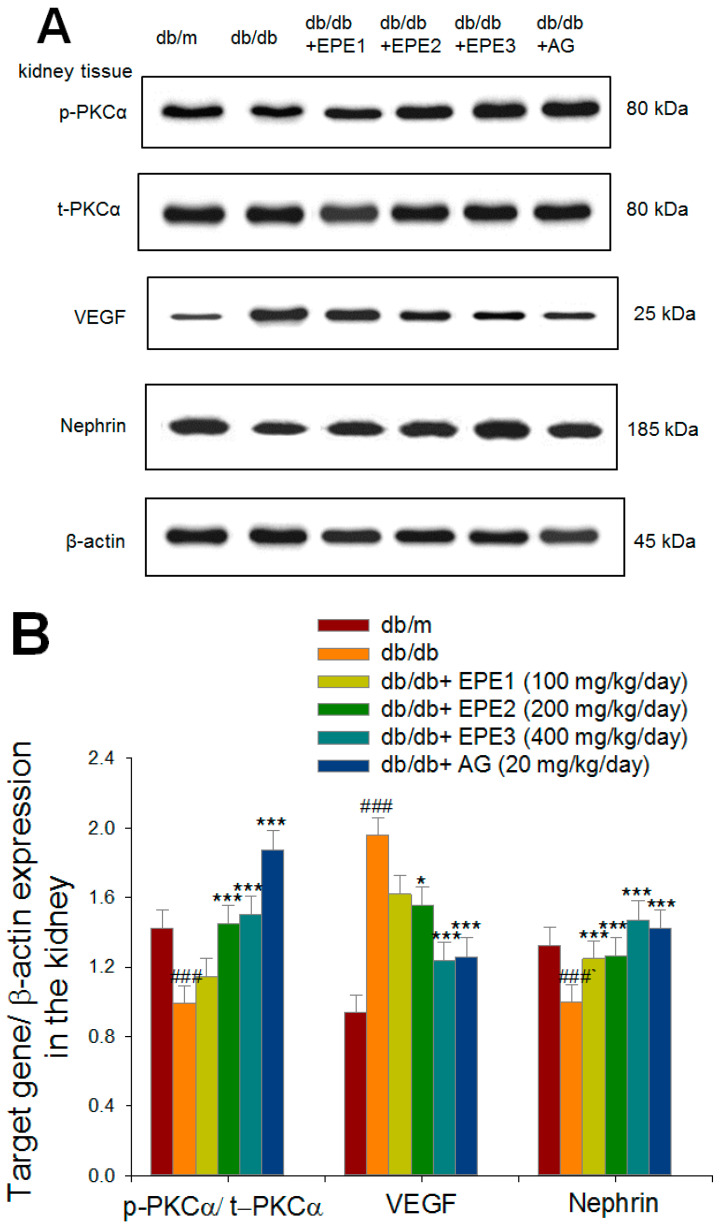
The kidney target gene expression levels in db/db mice following treatment with ethyl acetate extract of *P. emblica* L. (EPE) or aminoguanidine (AG, 20 mg/kg body weight) by Western blotting analysis on p-PKCα/t-PKCα, VEGF, nephrin, TGFβ1, collagen IV, fibronectin, Smad4, p-Smad2/t-Smad2, and p-Smad3/t-Smad3. (**A**,**C**,**E**,**G**) Representative image; (**B**,**D**,**F**,**H**) quantification of p-PKCα/t-PKCα, VEGF TGFβ1, collagen IV, Smad4, p-Smad2/t-Smad2, p-Smad3/t-Smad3, p-NLRP3/t-NLRP3, ICAM, and Capalase-1 to β-actin. Protein was separated by 12% SDS-PAG. ^###^ *p* < 0.001 as compared to the db/m group; * *p* < 0.05, ** *p* < 0.01, *** *p* < 0.001 compared to the db/db plus vehicle (distilled water) (db/db) group. All values are means ± SE (*n* = 8 per group). EPE, ethyl acetate extract of *P. emblica* L. EPE: EPE1: 100, EPE2: 200, EPE3: 400 mg/kg body weight; AG: aminoguanidine (20 mg/kg body weight).

**Figure 6 ijms-25-06686-f006:**
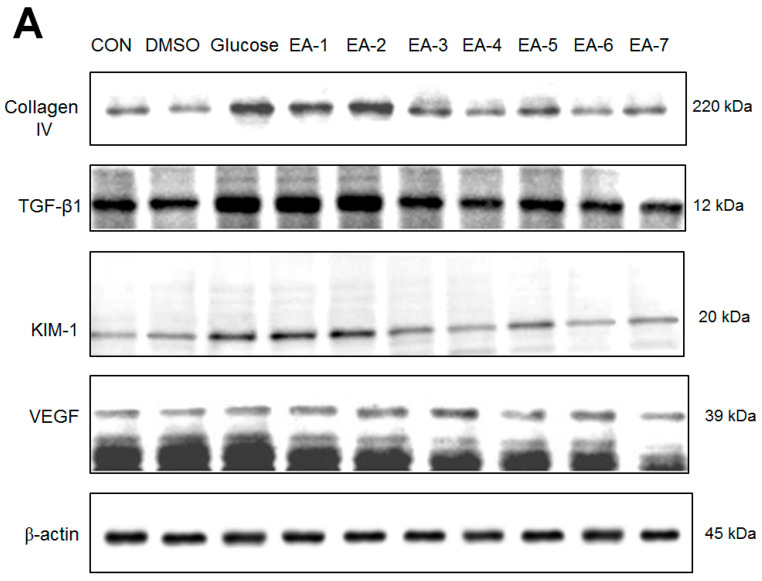
Effects of seven fractions of EPE (EA) on expression levels of collagen IV, TGF-β1, VEGF, and KIM in human renal mesangial (HRM) cells by Western blotting analyses. Human renal mesangial (HRM) cells were treated with seven fractions and equal amounts of lysates were resolved by SDS-PAGE and blotted for collagen IV, TGF-β1, VEGF, and KIM. (**A**–**E**) Effects of 7 fractions of ethyl acetate extract of *P. emblica* L. (EtOAc soluble fractions) (EA-1, EA-2, EA-3, EA-A, EA-5, EA-6, and EA-7) on expression levels of Collagen IV, TGF-β1, KIM, and VEGF in HRM cells by Western blotting analysis. Human renal mesangial (HRM) cells were treated with 7 fractions as described in the experimental procedures and equal amounts of lysates were resolved by SDS-PAGE and blotted for collagen IV, TGF-β1, KIM, and VEGF. (**A**) Representative blots for 7 fractions in HRM cells; (**B**–**E**) Quantification of the expression levels to β-actin including (**B**) collagen IV, (**C**) TGF-β1, (**D**) KIM, and (**E**) VEGF. All values are means ± S.E. ^###^ *p* < 0.001 compared with the control group; * *p* < 0.05, *** *p* < 0.001 compared with the glucose group. (**F**,**G**) High-performance liquid chromatography analysis of (**F**) 2500 ppm ethyl acetate of *P. emblica* L. (EPE), (**G**) 10.3 mg/5 mL EA-4 of ethyl acetate of *P. emblica* L.

**Figure 7 ijms-25-06686-f007:**
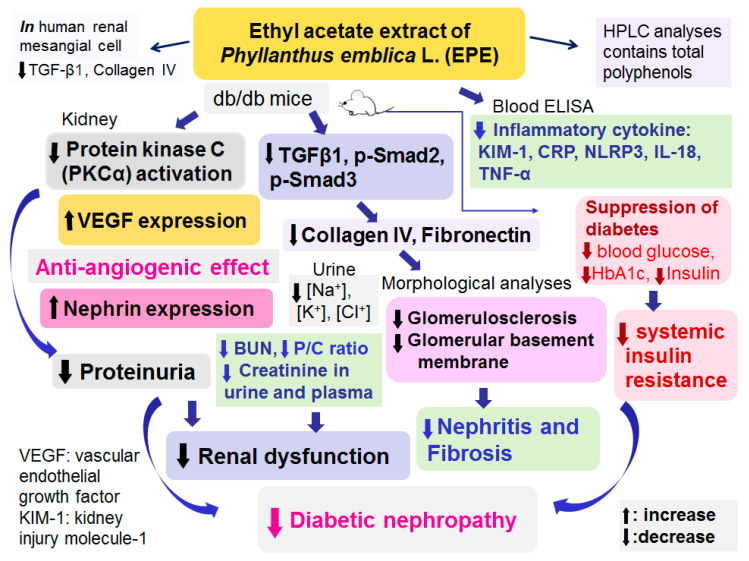
Graphic abstract of ethyl acetate extract of *P. emblica* L. (EPE) on diabetic nephropathy in a db/db mouse model.

**Table 1 ijms-25-06686-t001:** Effects of ethyl acetate extract of *P. emblica* L. (EPE) on relative organ weights (g).

Drug	Dose (mg/kg/day)	Liver Tissue (%)	Kidney Tissue (%)	Pancreas (%)	Skeletal Muscle (%)
db/m		5.8 ± 0.2	1.8 ± 0.2	0.5 ± 0.0	1.23 ± 0.08
db/db		4.9 ± 0.3 ^###^	0.8 ± 0.1 ^###^	0.6 ± 0.1	0.46 ± 0.08 ^###^
db/db + EPE1	0.1	4.6 ± 0.2	0.7 ± 0.0	0.5 ± 0.1	0.38 ± 0.04
db/db + EPE2	0.2	4.6 ± 0.2	1.0 ± 0.1	0.7 ± 0.0	0.52 ± 0.03
db/db + EPE3	0.4	4.2 ± 0.2	0.9 ± 0.0	0.6 ± 0.1	0.51 ± 0.03
db/db + AG	0.02	3.8 ± 0.3 *	0.8 ± 0.1	0.3 ± 0.1 *	0.48 ± 0.05

All values are means ± S.E. (*n* = 8). ^###^ *p* < 0.001 compared with the db/m group; * *p* < 0.05 compared with the db/db + vehicle (distilled water) (db/db) group. EPE, ethyl acetate extract of *P. emblica* L. EPE: EPE1: 100, EPE2: 200, EPE3: 400 mg/kg body weight; AG: aminoguanidine (20 mg/kg body weight).

## Data Availability

Where no new data were created, or where data is unavailable due to privacy or ethical restrictions. All data used to support the findings of this study are available from the corresponding author, Chun-Ching Shih, upon reasonable request. Corresponding author’s email: ccshih@ctust.edu.tw.

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
