# Peer review of "The Ethyl Acetate Extract of Phyllanthus emblica L. Alleviates Diabetic Nephropathy in a Murine Model of Diabetes"

_ijms, 2024, doi:10.3390/ijms25126686_

Round 1

Reviewer 1 Report

Comments and Suggestions for Authors

In the manuscript entitled "Protective Effects of Phyllanthus emblica L. Extract Against Renal Dysfunction, Angiogenesis, and Fibrosis Occur through the Regulation of PKCα Activation, VEGF, and TGF-β1 in a Diabetic Nephropathy Model" Cheng-Hsiu Lin and Chun-Ching Shih describe a series of in vitro and in vivo experiments by which they suggest that an ethyl acetate extract of P. emblica (EPE) alleviates diabetic nephropathy in a murine model of diabetes.

Overall, the study presents significant contributions to the field. I consider that this work could be valuable for publication in the magazine. However, I have some minor observations regarding the submitted manuscript.

·       Graphs A (body weight) and B (blood glucose levels) in Figure 2 are difficult to interpret. They could improve their visual presentation by using different colors for each comparison group and increasing the size of the graph.

·       In Figure 3C, corresponding to immunohistochemical staining for nephrin, the authors show histological kidney sections. They describe that db/db mice treated with EPE1, EPE2, EPE3, and AG had higher levels of nephrin expression compared with those in the db/db group. However, a quick interpretation might suggest that db/db, db/db+EPE1, and db/db+AG mice have similar nephrin expression. Therefore, it is suggested that the authors perform a relative quantification of the nephrin-positive area in the immunohistochemical stains to obtain a better interpretation of the results.

·       The results presented in Table 1 suggest that the GA-treated db/db mice had reduced relative weights of liver and pancreas tissue compared with those of the db/db group. The results of the experiment showed that there were no differences in the relative weights of liver tissue, kidney tissue, pancreas and skeletal muscle between db/db mice treated with EPE1, EPE2 and EPE3 and the db/db group. What do these results indicate? What is the correct interpretation? db/db mice normally have reduced relative weights compared to controls, are treatments for diabetes and diabetic nephropathy expected to restore the normal weight of these organs? The authors could address this information in the discussion section.

Author Response

Dear Reviewer 1:

Outline the changes made:

Reviewer: 1

  1. As Reviewer suggests, “In the manuscript entitled "Protective Effects of Phyllanthus emblica L. Extract Against Renal Dysfunction, Angiogenesis, and Fibrosis Occur through the Regulation of PKCα Activation, VEGF, and TGF-β1 in a Diabetic Nephropathy Model" Cheng-Hsiu Lin and Chun-Ching Shih describe a series of in vitro and in vivo experiments by which they suggest that an ethyl acetate extract of P. emblica (EPE) alleviates diabetic nephropathy in a murine model of diabetes. ”

And revised into

----> Title: The Ethyl Acetate Extract of Phyllanthus emblica L. Alleviates Diabetic Nephropathy in a Murine Model of Diabetes

  1. As Reviewer suggests, “Graphs A (body weight) and B (blood glucose levels) in Figure 2 are difficult to interpret. They could improve their visual presentation by using different colors for each comparison group and increasing the size of the graph.”

And revised into

----> Figure 2

  1. As Reviewer suggests, “In Figure 3C, corresponding to immunohistochemical staining for nephrin, the authors show histological kidney sections. They describe that db/db mice treated with EPE1, EPE2, EPE3, and AG had higher levels of nephrin expression compared with those in the db/db group. However, a quick interpretation might suggest that db/db, db/db+EPE1, and db/db+AG mice have similar nephrin expression. Therefore, it is suggested that the authors perform a relative quantification of the nephrin-positive area in the immunohistochemical stains to obtain a better interpretation of the results.”

And revised into

---->

IHC staining revealed that EPE2- and EPE3-treated db/db mice had increased nephrin expression levels and Nephrin-positive area (%) score by IHC stains compared with those of the db/db group (Figure 4C, D).

Figure 4. (C) expression levels of nephrin in the kidneys by immunohistochemical (IHC) staining; (D) Nephrin-positive area (%) score by IHC stains compared with the db/db group. EPE: EPE1, EPE2, and EPE3 (100, 200, and 400 mg/kg body).

  1. As Reviewer suggests, “The results presented in Table 1 suggest that the GA-treated db/db mice had reduced relative weights of liver and pancreas tissue compared with those of the db/db group. The results of the experiment showed that there were no differences in the relative weights of liver tissue, kidney tissue, pancreas and skeletal muscle between db/db mice treated with EPE1, EPE2 and EPE3 and the db/db group. What do these results indicate? What is the correct interpretation? db/db mice normally have reduced relative weights compared to controls, are treatments for diabetes and diabetic nephropathy expected to restore the normal weight of these organs? The authors could address this information in the discussion section. “

And revised into

----> The results presented in Table 1 suggest that the AG-treated db/db mice had reduced relative weights of liver and pancreas tissue compared with those of the db/db group. The results of the experiment showed that there were no differences in the relative weights of liver tissue, kidney tissue, pancreas and skeletal muscle between db/db mice treated with EPE1, EPE2 and EPE3 and the db/db group. db/db mice normally have reduced relative weights compared to controls, and EPE1, EPE2 and EPE3 treatments for diabetes and diabetic nephropathy expected to restore the normal weight of these organs, but not in the present results. An explanation for the abnormality of EPE treatments may be due to regulation of PCKα and downregulating expression levels of VEGF, thus contributing to a decrease in proteinuria, and in turn, a reduction in kidney hypertrophy to improve renal function, and finally reduce the progress rate of DN, thus there is no difference in relative weights of kidney tissue between EPE1, EPE2, and EPE3 treatments and db/db mice. Another explanation for the abnormality may represent complex physiological phenomena among peripheral tissues in murine body not easily to be interpreted.

  1. As Reviewer suggests, “Comments on the Quality of English Language

Language - I am not qualified to assess the quality of English in this paper.“

----> (The manuscript had already improved by American Journal of Editing (AJE) Order ID: SMMJZZL1M) (Services: - Standard Editing and - Presubmission Review) in February 25 2024) We could improve on the Quality of English Language by IJMS Journal assistants and I will pay, and nevertheless at the present because other Reviewer’s comments are too many points to improve this point in time, please give me additional time, thank you.

  1. Recently, our submitted report (submitted Ecam manuscript ID:7802863 was withdraw since the Ecam publisher was closed and withdraw all submitted manuscripts including my manuscript, and did not have any process and all articles were switched to the WILEY and did not have any methods to help us)

And revised into

---->The present HPLC analysis showed the main ingredient of EA-4 was chebulagic acid (therefore we add Figure 6F-G in this manuscript).

Reviewer 2 Report

Comments and Suggestions for Authors

The study describes the nephropathy protection by Phyllanthus emblica fruit extracts in diabetic mouse. 

The study is very exhaustive and currently does not look to have been organized. Authors can try to organize the experiments under specific objectives and then comprehend their findings. 

The article requires a meticulous English correction. Several words are inappropriately used and that is misleading the reader. For example, line 111- it is currently meaning that that study was looking how EPE treatment worsened neprophathy (improved renal dysfunction or improved renal function). Similarly, line 455, it should be voucher specimen were deposited. Voucher specimen were replaced gives an irrelevant meaning.

There are several sentences like line 299 which has open ended meaning and redundant. Introduction need to be made precise, currently it is giving lot of information like a review. Also, I think figure 1 need to re-worked. Picture of fruits can be separated from the other western blot panel. 

May be authors can try to separate the manuscript into two- one with the mouse study and another with the invitro studies. Currently there is too much of information which looks repeating without any comprehension. The abstract itself is very lengthy to grasp the whole study. 

Comments on the Quality of English Language

Extensive English language editing is required.

Author Response

Dear Reviewer 2:

Outline the changes made:

Reviewer: 2

  1. As Reviewer suggests, “The article requires a meticulous English correction. Several words are inappropriately used and that is misleading the reader. For example, line 111- it is currently meaning that that study was looking how EPE treatment worsened nephropathy (improved renal dysfunction or improved renal function).”

And revised into

----> line 111th : At present, the major clinical treatment method for diabetic nephropathy is to control glycemia and hypertension (such as the clinic drug: angiotensin II converting enzyme inhibitor; ACEI) and restriction of food protein intake and this method could not control disease progress from the beginning to the end; nevertheless, the therapeutic effectiveness of this method is limited, and the only way to slow the progression of diabetic nephropathy is and improve renal function and reduce both the incidence of the end-stage of DN and the proteinuria to provide patients with good control.

  1. As Reviewer suggests, “Similarly, line 455, it should be voucher specimen were deposited. Voucher specimen were replaced gives an irrelevant meaning”

And revised into

----> line 451~455: These fruits were identified by Department of Chinese Pharmaceutical Sciences and Chinese Medicine Resources of China Medical University, Taiwan, where the voucher specimen (CMPF385) were deposited. Fruits of P. emblica L. (Figure 1) were purchased from Taiwan Miaoli County, Ogan Marketing Cooperation. A total of 7.3 kg of fruits of Phyllanthus emblica L. was dried and pulverized into 1.34 kg of fine powder.

  1. As Reviewer suggests, “There are several sentences like line 299 which has open ended meaning and redundant.”

And revised into

----> line 299 Urinary albumin excretion may be a predictive factor for the prognosis of DN, and urinary albumin indicates impaired renal function [8,9], and the results of these our studies demonstrated that EPE treatment decreased urinary albumin is associated with and displayed a kidney-protective effect in type 2 diabetes mellitus. Since the postulated functional role of the classical PKC isoforms activation and the signal pathway including VEGF, nephrin, and proteinuria plays in the development of diabetic nephropathy [17−21], the present study was designed to clarify whether EPE ameliorates diabetic nephropathy and reduces proteinuria by PKC isoforms activation and regulation of target genes including VEGF and nephrin expressions.

  1. As Reviewer suggests, “The study currently does not look to have been organized. Authors can try to organize the experiments under specific objectives and then comprehend their findings. Introduction need to be made precise, currently it is giving lot of information like a review”

And revised into

  1. ----> Introduction

Diabetic nephropathy (DN) is a serious and common complication of diabetes mellitus. Diabetic nephropathy is a morbid microvascular complication associated with diabetes [1]. Diabetic nephropathy is the leading cause of end-stage kidney disease worldwide [2] and is an independent risk factor for all-cause and cardiovascular mortality in diabetic patients [3]. Approximately 30-40% of patients with type 2 diabetes develop diabetic nephropathy [4]. Diabetic nephropathy is the main complication of type 2 diabetes and leads to glomerular membrane thickening, mesangial expansion, glomerular hypertrophy, and overt renal disease [5,6,7].

The markers of renal function include urinary albumin excretion, plasma creatinine, urine creatinine, the urine albumin creatine ratio (UACR; the urine P/C ratio), and histological changes within the kidney structures. Blood urea nitrogen (BUN) is a protein metabolite product. Creatinine is a muscle metabolite product. High BUN and creatinine levels indicate renal dysfunction. Urinary albumin excretion indicates impaired renal function [8,9]. Persistent albuminuria is a hallmark of DN.

The postulated functional role of the classical protein kinase C (PKC) isoforms and the signal pathway via vascular endothelial growth factor (VEGF), nephrin, and then proteinuria, which played a core role in the development of diabetic nephropathy [10-14]. Protein kinase C (PKC) activation can directly increase the permeability of albumin and other macromolecules through barriers formed by endothelial cells [10,11]. Vascular endothelial growth factor (VEGF) participates in the regulation of glomerular permeability, glomerular endothelial cell growth, and urinary albumin excretion [12,13,14].

Evidence has emphasized the critical role of inflammation in the pathogenesis of DN [15]. The pathogenesis mechanisms of DN are very complicated and remain unclear. In diabetes mellitus, hyperglycemia induces the activation of various signaling pathways involved in diabetic vascular complications through the production of inflammatory cytokines. Renal inflammation and subsequent fibrosis are critical processes leading to end-stage DN [16,17]. Renal inflammation and subsequent fibrosis are critical processes leading to end-stage DN [16,17].

In DN, the accumulation of extracellular matrix (ECM) components in the glomerular mesangium and tubulointerstitium causes early glomerular hypertrophy and eventually glomerulosclerosis and tubulointerstitial fibrosis [18]. Collagen IV accumulation is a crucial phenomenon underlying mesangial expansion [19]. Transforming growth factor-β (TGF-β) is a key regulator of fibrosis that promotes the accumulation of ECM. TGF-β induces phosphorylation and activation of Smad signaling pathway components [20]. TGF-β1 is another important factor in the pathogenesis of DN and mediates the inflammatory response, which exacerbates ECM secretion involving fibronectin and collagen accumulation and accelerates glomerulosclerosis in diabetes [21].

db/db mice serve as an animal model of obesity-related diabetes and can be used to study kidney changes due to diabetes [22]. db/db mice are overweight, hyperglycemic and hyperinsulinemic and exhibit increased kidney weight, glomerular mesangial matrix and albumin excretion [23]. The concentration of collagen IV increases with DN progression in patients and db/db mice [24].

  1. As Reviewer suggests, “introduction provide sufficient background and include all relevant references”

And revised into

----> Phyllanthus emblica L. (Figure 1) is a plant that is widely distributed in tropical and subtropical regions such as Taiwan. The fruit of P. emblica possesses various pharmacological activities. Recently, our findings have shown that Phyllanthus emblica extract displays an antihyperglycemic activity both in in NOD with spontaneous and cyclophosphamide-accelerated Type 1 diabetic mice [25] and in streptozotocin-induced Type 1 diabetic mice [26]. Triphala, a well-recognized and highly efficacious polyherbal Ayurvedic medicine consisting of fruits of the plant species Emblica officinalis (Amalaki), Terminalia bellerica (Bibhitaki), and Terminalia chebula (Haritaki), is a cornerstone of gastrointestinal and rejuvenative treatment [27]. Recently, Triphala is demonstrated to ameliorate nephropathy via inhibition of TGF-β1 and oxidative stress in diabetic rats [28]. Nevertheless, Triphala is consisting of fruits of the three plant species, but not merely Phyllanthus emblica L. extract exhibits an ameliorative effect on diabetic nephropathy. More recently, Emblica officinalis fruit extract is reported to display nephroprotective effects against malachite green toxicity by restoration of the renal cytoarchitecture and enhanced activity of antioxidant enzymes in piscine model [29], and these findings are in vitro study. However, the ameliorative activity of ethyl acetate extract of P. emblica (EPE) is not well defined in db/db mice. Therefore, the present study was designed to clarify EPE’s effects on Type 2 diabetes and DN in an animal model of db/db mouse.

  1. As Reviewer suggests, “The study currently does not look to have been organized. Authors can try to organize the experiments under specific objectives and then comprehend their findings.”

And revised into

---->At present, the major clinical treatment method for DN is to control glycemia and improve renal function and reduce both the incidence of the end-stage of DN and the proteinuria to provide patients with good control. The present study is consisted of three parts: whether EPE could regulate target genes expressions including TGF-β1 and collagen IV in vitro or not, the protective activity and mechanism of EPE on DN by a db/db mouse model, and to clarify the responsible components for the amelioration of DN, we examined the target genes expressions ameliorating of DN of EPE by Western blotting test in vitro and then HPLC analysis of refractions of EPE were performed.

The animal study was designed to evaluate whether EPE could improve insulin resistance as well as ameliorate DN by the improvement of renal dysfunction (such as albumin excretion) and ameliorative glomerular morphological damage by immunohistochemistry or not, and to examine the expression levels of targeted genes (including the expression levels of p-PKCα/ t-PKCα and VEGF, and renal fibrosis biomarkers including TGF-β1, collagen IV, p-Sma and mad protein (Smad)2, p-Smad3, and Smad4) by Western blotting. Finally, this study examined the effects of seven refractions of EPE on the target expression levels of Collagen IV, TGF-β1, VEGF, and inflammatory cytokine kidney injury molecule (KIM)-1 in Human Renal Mesangial (HRM) cells to clarify the main ingredients responsible for the protective effects of EPE against diabetic nephropathy from the early stage to the end stage including inflammation, fibrosis, and angiogenesis.

[27] Peterson, C.T.; Denniston, K.; Chopra, D. Therapeutic Uses of Triphala in Ayurvedic Medicine. J Altern Complement Med. 2017, 23(8), 607–614.

[28] Suryavanshi, S.V.; Garud, M.S.; Barve, K.; Addepalli, V.; Utpat, S.V.; Kulkarni, Y.A. Triphala ameliorates nephropathy via inhibition of TGF-β1 and oxidative stress in diabetic rats. Pharmacology 2020, 105(11-12), 681691.

[29] Sinha, R.; Jindal, R.; Faggio, C. Nephroprotective effect of Emblica officinalis fruit extract against malachite green toxicity in piscine model: Ultrastructure and oxidative stress study. Micro. Res. Tech. 2021, 84(8), 1911–1919.

  1. As Reviewer suggests, “Also, I think figure 1 need to re-worked. Picture of fruits can be separated from the other western blot panel.”

And revised into

----> Figure 1

Figure 1. Fruits of Phyllanthus emblica L..

Figure 2

Figure 2. The targeted genes effects of Ethyl acetate extract of Phyllanthus emblica L. (EPE) in human renal mesangial cell lysates by Western blot analysis with a primary antibody against TGF-β1 and Collagen IV. (A) Representative blots in human renal mesangial cells; (B~C) Quantification of the expression levels of TGF-β1 and Collagen IV. All values are means ± S.E. +++ p < 0.001 compared with the control group; * p < 0.05, *** p < 0.001 compared with the HG group. The β-actin was used as the internal standard in each sample. HG: high glucose. EPE: Ethyl acetate extract of Phyllanthus emblica L.

Figure 2. ----> Figure 3.

Figure 3. ----> Figure 4.

Figure 4. ----> Figure 5.

Figure 5. ----> Figure 6.

Figure 6. ----> Figure 7.

  1. As Reviewer suggests, “The abstract itself is very lengthy to grasp the whole study.”

And revised into

---->

Oil-Gan, also known as emblica, is the fruit of the genus Phyllanthus emblica L. The fruits of these trees are high in nutrients and have excellent effects on health care and development values. There are many methods for the management of diabetic nephropathy (DN). However, there is a lack of effective drugs for treating DN from throughout the disease course. The primary aim of this study was to examine the protective effects (including urine and blood analyses and the measurement of inflammatory cytokine levels, and histological examination) and mechanisms of action of an ethyl acetate extract of P. emblica (EPE) on db/db mice, an animal model of diabetic nephropathy, as well as a secondary aim of examining the expression levels of p- protein kinase Cα (PKCα)/t-PKCα in the kidney and its downregulation of vascular endothelial growth factor (VEGF) and fibrosis gene transforming growth factor-β1 (TGF-β1) by Western blot analyses. Eight db/m C57BLKS/J mice were used as the control group. Forty db/db mice were randomly divided into five groups. Treatments included vehicle, EPE1, EPE2, EPE3 (at doses of 100, 200, or 400 mg/kg EPE), or the comparative drug aminoguanidine for 8 weeks. After 8 weeks of treatment, blood and urine samples were collected for analyses of blood glucose, insulin, and HbA1c levels, and pancreatic and kidney tissues were dissected for histological analyses. The present study indicated that the administration of EPE to db/db mice effectively controlled hyperglycemia and hyperinsulinemia by markedly lowering blood glucose, insulin, and glycosylated HbA1c levels. Administration of EPE to db/db mice decreased the levels of BUN and creatinine both in blood and urine and reduced urinary albumin excretion and the albumin creatine ratio (UACR) in urine. Moreover, EPE treatment decreased the blood levels of inflammatory cytokines, including kidney injury molecule-1 (KIM-1), C-reactive protein (CRP), and NLR family pyrin domain containing (NLRP3). Our findings showed that EPE not only had antihyperglycemic effects but also improved renal function in db/db mice. Histological examination of the kidney by immunohistochemistry indicated that EPE can improve kidney function by ameliorating glomerular morphological damage following glomerular injury; alleviating proteinuria by upregulating the expression of nephrin, a biomarker of early glomerular damage; and inhibiting glomerular expansion and tubular fibrosis. Moreover, the administration of EPE to db/db mice increased the expression levels of p- PKCα/t-PKCα but decreased the expression levels of VEGF and renal fibrosis biomarkers (TGF-β1, collagen IV, p-Smad2, p-Smad3, and Smad4), as shown by Western blot analyses. These results implied that EPE as a supplement has a protective effect against renal dysfunction through the amelioration of insulin resistance as well as the suppression of nephritis and fibrosis in a DN model.

  1. Recently, our submitted report (submitted Ecam manuscript ID:7802863 was withdraw since the Ecam publisher was closed and withdraw all submitted manuscripts including my manuscript, and did not have any process and all articles were switched to the WILEY and did not have any methods to help us)

And revised into

---->The present HPLC analysis showed the main ingredient of EA-4 was chebulagic acid (therefore we add Figure 6F-G in this manuscript).

  1. As Reviewer suggests, “Comments on the Quality of English Language should be improved. Extensive editing of English language required “

----> (The manuscript had already improved by American Journal of Editing (AJE) Order ID: SMMJZZL1M) (Services: - Standard Editing and - Presubmission Review) in February 25 2024) We could improve on the Quality of English Language by IJMS Journal assistants and I will pay, and nevertheless at the present because other Reviewer’s comments are too many points to improve this point in time, please give me additional time, thank you.
